# GFCNet: Contrastive Learning Network with Geography Feature Space Joint Negative Sample Correction for Land Cover Classification

**Zhaoyang Zhang [1,2], Wenxuan Jing [1], Haifeng Li [1,2], Chao Tao [1] and Yunsheng Zhang [1,2,*]**

[1] School of Geosciences and Info-Physics, Central South University, Changsha 410083, China; zhaoyangzhang@csu.edu.cn (Z.Z.); 215012162@csu.edu.cn (W.J.); lihaifeng@csu.edu.cn (H.L.); kingtaochao@csu.edu.cn (C.T.)

[2] Xiangjiang Laboratory, Changsha 410205, China

* Correspondence: zhangys@csu.edu.cn

**Abstract:** With the continuous improvement in the volume and spatial resolution of remote sensing images, the self-supervised contrastive learning paradigm driven by a large amount of unlabeled data is expected to be a promising solution for large-scale land cover classification with limited labeled data. However, due to the richness and scale diversity of ground objects contained in remote sensing images, self-supervised contrastive learning encounters two challenges when performing large-scale land cover classification: (1) Self-supervised contrastive learning models treat random spatial–spectral transformations of different images as negative samples, even though they may contain the same ground objects, which leads to serious class confusion in land cover classification. (2) The existing self-supervised contrastive learning models simply use the single-scale features extracted by the feature extractor for land cover classification, which limits the ability of the model to capture different scales of ground objects in remote sensing images. In this study, we propose a contrastive learning network with Geography Feature space joint negative sample Correction (GFCNet) for land cover classification. To address class confusion, we propose a Geography Feature space joint negative sample Correction Strategy (GFCS), which integrates the geography space and feature space relationships of different images to construct negative samples, reducing the risk of negative samples containing the same ground object. In order to improve the ability of the model to capture the features of different scale ground objects, we adopt a Multi-scale Feature joint Fine-tuning Strategy (MFFS) to integrate different scale features obtained by the self-supervised contrastive learning network for land cover classification tasks. We evaluate the proposed GFCNet on three public land cover classification datasets and achieve the best results compared to seven baselines of self-supervised contrastive learning methods. Specifically, on the LoveDA Rural dataset, the proposed GFCNet improves 3.87% in Kappa and 1.54% in mIoU compared with the best baseline.

**Keywords:** land cover classification; contrastive learning; self-supervised learning; remote sensing image understanding

## 1. Introduction

As an important research topic in the field of Earth observation (EO), large-scale land cover classification plays an important role in many studies and applications, such as urban expansion monitoring, resource and environmental protection, socioeconomic assessment, etc. [1–3]. Traditional remote sensing image land cover classification methods such as Support Vector Machine (SVM) and Random Forest (RF) rely on manual feature extraction and rule formulation, which promotes the development of land cover classification [4–6].

In contrast, data-driven deep learning techniques that have emerged since then have performed well in land cover classification and further improved accuracy [3,7,8]. Most of

these algorithms are based on the supervised learning paradigm, which constructs supervised signals (usually a loss function calculated using labeled data and model predictions) through high-quality labeled data [9–14].

In recent years, with the continuous improvement in the volume and spatial resolution of remote sensing images covering the world [15], compared with supervised learning models that use high-cost labeled data as supervised signals, self-supervised contrastive learning models driven by a large number of unlabeled data are expected to become an effective solution for large-scale land cover classification with limited labeled data [16–20].

Compatible with the spatiotemporal invariance of remote sensing images, the self-supervised contrastive learning model constructs positive and negative samples (usually remote sensing image patches used for deep learning training) based on the basic understanding that the semantic information of the same image does not change with spectral and geometric distortions [21–26], and has achieved competitive results in remote sensing image scene classification [22,23,27], semantic segmentation [25,28,29], and object detection [30,31].

However, due to the richness and scale diversity of ground objects contained in remote sensing images, the self-supervised contrastive learning model encounters two challenges in land cover classification [25,27–29,32]. The first challenge is the ground object class confusion. The self-supervised contrastive learning model adopts a heuristic positive and negative sample construction strategy due to the lack of available label data [21,33]. The random spatial–spectral transformations of the same image are regarded as positive samples, and the random spatial–spectral transformations of different images are regarded as negative samples [21,34–36] (positive samples are pulled closer while the negative samples are pushed farther away). As a result, when the model deals with remote sensing images containing rich and complex ground objects, it inevitably leads to the existence of images containing the same objects in the negative sample pairs [25,37]. A natural way to address this challenge is to introduce clustering methods to reduce the risk of negative samples containing the same ground object [38–40], but due to the intra-class variability and inter-class similarity of ground objects in remote sensing images [41], clustering methods may introduce additional class confusion when constructing positive and negative samples. In addition, some self-supervised contrastive learning methods consider using the self-correction ability of the model to eliminate the influence of negative samples containing the same ground objects on the model [25,33,37]. These methods effectively alleviate the class confusion, but they only use the feature space relationship of the image, and do not use the geospatial relationship of the image, although the geospatial relationship is easy to obtain for remote sensing images. The second challenge is that the existing self-supervised contrastive learning models simply use the single-scale features extracted by the feature extractor for land cover classification tasks [16–19], which limits the ability of the model to capture different scales of ground objects. To address this challenge, the existing self-supervised contrastive learning models consider adding local contrastive modules [28] or dense contrastive modules [29,42] in the pretraining stage of feature extraction. These methods effectively improve the performance of self-supervised contrastive learning for land cover classification tasks, but inevitably lead to higher computational overhead.

In this study, we propose a contrastive learning Network with Geography Feature space joint negative sample correction (GFCNet) for land cover classification. To address the serious ground object class confusion, we develop a Geography Feature space joint negative sample Correction Strategy (GFCS). Different from simply treating the data augmentation of different images as negative samples, we comprehensively consider the visual feature relationship and geospatial relationship of remote sensing images in the process of constructing negative samples. Specifically, we merge the geospatial distance and feature space distance of the image to obtain the geography feature space joint distance, and based on this, we screen the negative samples that contain the same ground object and correct these samples as positive samples to reduce the ground object class confusion. In order to improve the ability of the model to capture features of different scales of ground objects,

we adopt a Multi-scale Feature joint Fine-tuning Strategy (MFFS) to integrate the features at different scales obtained by the self-supervised contrastive learning model for land cover classification.

We compare the performance of GFCNet with seven self-supervised contrastive learning baseline methods on three public land cover classification datasets. The experimental results indicated that on the Five-Billion-Pixel dataset, the proposed GFCNet improves the Overall Accuracy (OA) by 0.56%, Kappa by 0.61%, and the mean Intersection-over-Union (mIoU) by 0.43% compared with the best baseline. On the LoveDA Urban dataset, the proposed GFCNet improves the OA by 1.14%, Kappa by 1.63%, and the mIoU by 1.39% compared with the best baseline. On the LoveDA Rural dataset, the proposed GFCNet improves the OA by 2.57%, Kappa by 3.84%, and the mIoU by 1.54% compared with the best baseline.

The main contributions of this paper are summarized as follows:

(1) We proposed a Geography Feature space joint negative sample Correction Strategy (GFCS), which comprehensively considers the geography space relationship and feature space relationship of the image to construct negative samples, effectively alleviating the class confusion of the self-supervised contrastive learning model for land cover classification.

(2) We utilized the Multi-scale Feature joint Fine-tuning Strategy (MFFS) to integrate the features of different scales obtained by the self-supervised contrastive learning model, which enhances the ability of the model to capture objects of different scales.

(3) Experimental results on three public land cover classification datasets indicated that the proposed GFCNet achieves the best results in all three metrics, OA, Kappa, and mIoU, compared to the baseline of seven self-supervised contrastive learning methods. In addition, GFCNet achieves a maximum improvement of 5.7% in Kappa and a maximum improvement of 4.85% in mIoU compared to the self-supervised contrastive learning methods with the original positive negative sample construction strategy.

## 2. Materials and Methods

### 2.1. Related Work

#### 2.1.1. Negative Sample Construction Strategy for Self-Supervised Contrastive Learning

The selection of negative samples plays an important role in self-supervised contrastive learning [43–46]. The original self-supervised contrastive learning methods treat the data augmentation (random geometric and spectral transformations of image patches) of different images as negative samples [21,34,35,47], but for remote sensing images that contain rich ground objects, this approach inevitably leads to ground object class confusion as the negative samples may also contain the same ground object [25,33,37]. To address the ground object class confusion, existing methods can be divided into three types.

The first type of method considers the introduction of unsupervised clustering to assist the construction of negative samples, such as DeepCluster [40], PCL [38], and SwAV [39]. These methods improve the construction quality of negative samples to a certain extent, but due to the richness and complexity of ground objects in remote sensing images, the error of the clustering method leads to additional class confusion in the model [25,37].

The second type of method considers not constructing negative samples to avoid the negative impact of negative samples containing the same ground object on the model, such as Barlow Twins [48] and BYOL [49]. However, this also means that the image invariance learned by the model is only derived from positive samples obtained by data augmentation, and the ability of the model to capture features of the ground object is more limited by the data augmentation methods [21,50–52].

The third type of method considers the correction of negative samples by the self-determination ability of the model, such as FALSE [25], FNC [33], and IFND [37], which do not introduce additional class confusion, but the construction quality of negative samples is limited by the self-determination ability of the model.

The above methods alleviate the class confusion to some extent, but they only consider the relationship between images in the feature space and ignore the relationship between images in the geography space. The geospatial relationship of the image is easy to obtain for remote sensing images, and this relationship can be easily used to infer the relationship between two neighboring image patches, such as up-sampling [53] and local interpolation of images.

### 2.1.2. Self-Supervised Contrastive Learning for Land Cover Classification

Early land cover classification relied on field surveys and visual interpretation of aerial or satellite images [54]. With the rapid development of satellite remote sensing technology and digital image processing technology, land cover classification has entered a new stage [7,55]. The introduction of machine learning methods, such as Support Vector Machine (SVM) and Random Forest (RF), has further promoted the development of remote sensing image land cover classification [4–6].

Later, the rise of deep learning technology has had a profound impact on land cover classification [56–58]. Models based on Convolutional Neural Network (CNN) [59–61] and Transformer [62,63] have performed well in land cover classification and further improved the accuracy. Most of these algorithms are based on the supervised learning paradigm and need to provide a large amount of high-quality labeled data [17,64].

In recent years, self-supervised contrastive learning has been widely used as a powerful image representation extraction method for land cover classification with limited labeled data [16–19]. Different from supervised learning, the land cover classification model of self-supervised contrastive learning is divided into two stages [16,28]. The first stage aims to use a large amount of unlabeled image data to train the image feature extractor through the pretext task (tasks that do not use labeled data to train models before land cover classification tasks, such as contrastive learning tasks). Therefore, the design of the pretext task in the first stage is the key to capture image features by the self-supervised contrastive learning model [21,36]. In order to make the model have the ability to capture local features of images, IndexNet [29] and VADeR [32] consider adding a dense contrastive module, which makes the pretext task more suitable for the downstream land cover classification task, but it also inevitably introduces additional computational overhead.

The second stage aims to use the image features obtained by the feature extractor to perform a land cover classification task. In this stage, a small amount of labeled data is needed to fine-tune the feature decoder and finally obtain the land cover classification results [18,19]. In order to adequately train the feature decoder, GLCNet [28] uses the feature maps obtained by the feature decoder in the pretraining stage to perform local contrastive, which improves the performance of the model for land cover classification.

However, the current self-supervised contrastive learning model simply uses the single-scale features of the image obtained by the feature extractor for the land cover classification task, which limits the ability of the model to capture different scales of ground objects.

### 2.2. Method

### 2.2.1. Overview

The contrastive learning Network with Geography Feature Space Joint Negative Sample Correction (GFCNet) for Land Cover Classification consists of two main parts: (1) geography feature space joint negative sample correction contrastive pretraining and (2) multi-scale feature joint land cover classification fine-tuning. The overall framework of GFCNet is shown in Figure 1.

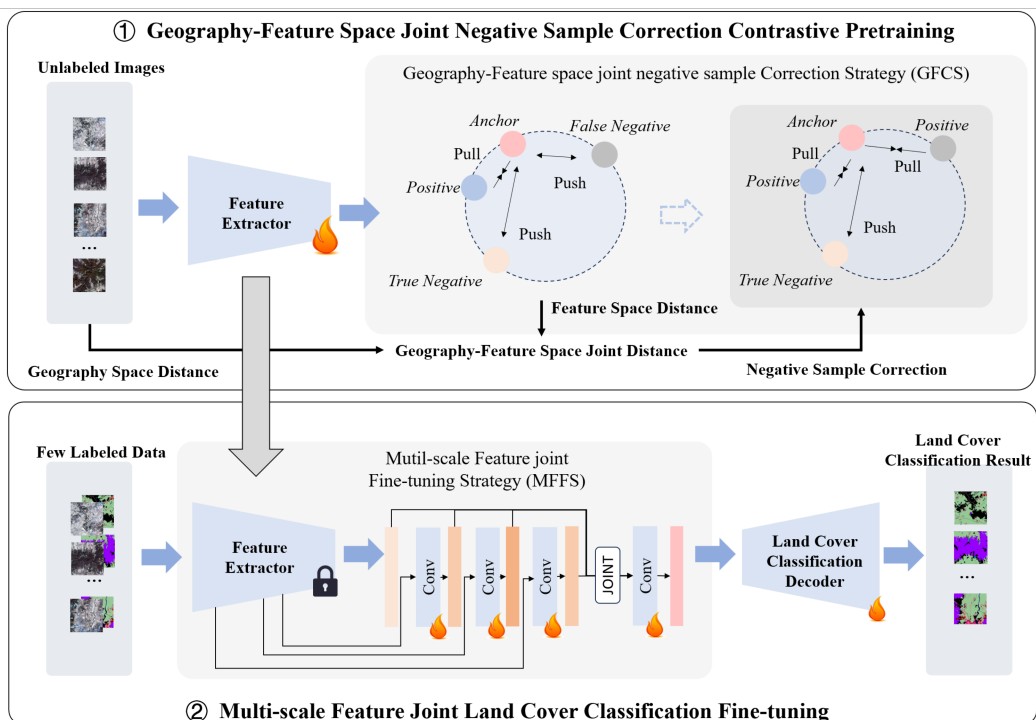

**Figure 1.** The overall framework of GFCNet, where the "lock" symbol in the lower right corner of the model indicates freezing network parameters and the "fire" symbol indicates updating network parameters. The first stage is the geography feature space joint negative sample correction contrastive pretraining: we synthesize the geospatial and feature space relationships of the image to construct the negative samples and update the feature extractor network parameters. The second stage is the multi-scale feature joint land cover classification fine-tuning: we freeze the feature extractor parameters obtained in the first stage, integrate the different scales of features obtained by the feature extractor, and update only the parameters of the single-layer convolutional that obtain multi-scale features and the land cover classification decoder.

The first part is corresponding to the unsupervised pretraining stage (the stage of training the image feature extractor using a large amount of unlabeled data) of self-supervised contrastive learning. In this stage, we adopt the proposed Geography Feature space joint negative sample Correction Strategy (GFCS), which integrates the geography space distance and feature space distance of remote sensing images, to improve the negative sample construction in the pretraining stage of self-supervised contrastive learning and alleviate the class confusion caused by images containing the same ground object in the negative sample pairs.

The second part is corresponding to the fine-tuning stage (the stage of training the whole model using a small amount of labeled data to obtain land cover classification results) of self-supervised contrastive learning. In this stage, we adopt a Multi-scale Feature Fine-tuning Strategy (MFFS) to integrate the different scale features obtained in the pretraining stage for downstream land cover classification tasks, and improve the ability of the model to capture different scales of ground objects.

### 2.2.2. Geography Feature Space Joint Negative Sample Correction Contrastive Pretraining

The geography feature space joint negative sample correction contrastive pretraining aims to comprehensively use the geography space and feature space relationship of images to construct negative samples, minimize the risk of negative sample images containing the same ground objects, and alleviate the class confusion of self-supervised contrastive learning (the visual analysis of negative sample correction is described in Section 3.4.4). Figure 2 illustrates the difference between SimCLR, FALSE, and GFCNet for constructing

positive and negative samples. It mainly consists of three parts: (1) geography feature space joint distance calculation, (2) negative sample determination and reconstruction, and (3) feature extraction and contrastive loss calculation.

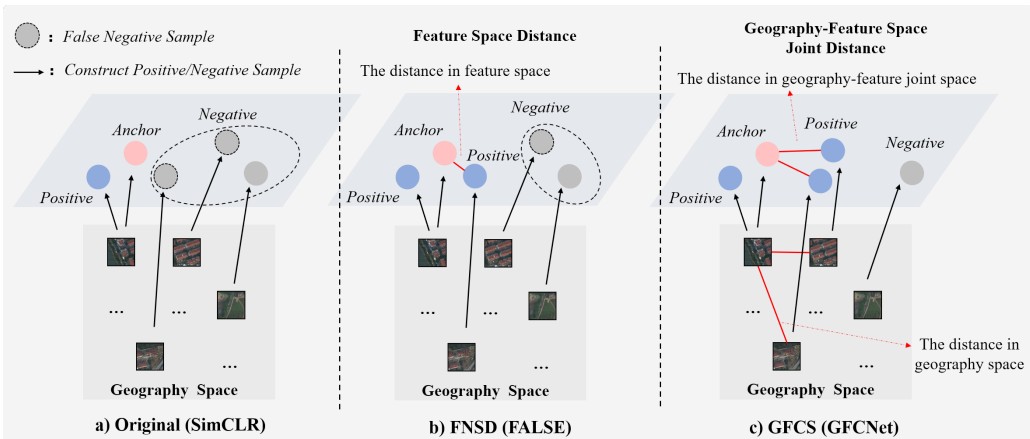

**Figure 2.** Diagram of the comparison between SimCLR, FALSE, and GFCNet for constructing positive and negative samples. (**a**) denotes the original positive and negative sample construction method adopted by SimCLR, in which all images different from the anchor sample are considered as negative samples; (**b**) denotes the False Negative Self-Determination (FNSD) adopted by FALSE, which corrects the false negative samples closer to the anchor sample in the feature space to positive samples; (**c**) denotes the proposed Geography Feature space joint negative sample Correction Strategy (GFCS) adopted by GFCNet, which indicates that the false negative samples closer to the anchor sample in geographic feature joint space are corrected as positive samples.

(1) Geography Feature Space Joint Distance Computation

We use the Euclidean distance of the image spatial index as the geography space distance of the image, the cosine similarity of the features as the feature space distance of the image, and use the normalized inverse geography space distance as the weight to integrate with the feature space distance to obtain the geography feature space joint distance.

Specifically, when calculating the geography space distance of images, we first establish a two-dimensional spatial index $(x, y)$ for each image based on the geospatial information. The two-dimensional spatial index records the relative spatial position of each image in the whole dataset. For example, for images $I_a$ and $I_b$, if their corresponding two-dimensional spatial indexes are $(x_a, y_a)$ and $(x_b, y_b)$, respectively, the difference of their two-dimensional spatial indexes $(x_a - x_b, y_a - y_b)$ records the number of images $|x_a - x_b|$ that are separated by the horizontal axis and the number of images $|y_a - y_b|$ that are separated by the vertical axis. Then, we compute the Euclidean distance between the two-dimensional spatial index of the two images as the geography space distance. The geography space distance $D_g$ of images $I_a$ and $I_b$ can be computed with the following equation:

$$D_g(I_a, I_b) = \sqrt{(x_a - x_b)^2 + (y_a - y_b)^2}.$$ (1)

When calculating the feature space distance of images, we first project the image to the feature space through the feature extractor. Then, we calculate the cosine similarity of the image features as the feature space distance of the image. Specifically, for images $I_a$ and $I_b$, if their corresponding features are $f_a$ and $f_b$, then the feature space distance $D_f$ of images $I_a$ and $I_b$ can be computed with the following equation:

$$D_f(I_a, I_b) = \frac{f_a \cdot f_b}{\|f_a\| \|f_b\|}.$$ (2)

Finally, on the basis of obtaining the image geography space distance and feature space distance, we integrate the normalized inverse geography space distance as a weight with

the feature space distance to obtain the geography feature space joint distance. Specifically, for images $I_a$ and $I_b$, if their geography space distance is $D_g(I_a, I_b)$ and their feature space distance is $D_f(I_a, I_b)$, the geography feature space distance $D_{gf}(I_a, I_b)$ can be computed by the following equation:

$$D_{gf}(I_a, I_b) = w(I_a, I_b) \cdot D_f(I_a, I_b),\tag{3}$$

$$w(I_a, I_b) = \begin{cases} 1 & D_g(I_a, I_b) = 0 \\ \frac{1/D_g(I_a, I_b)}{\Sigma_{i=1}^{n} 1/D_g(I_a, I_i)} & D_g(I_a, I_b) \neq 0 \end{cases},\tag{4}$$

where $n$ represents the number of images of a batch of the input model.

(2) Negative Sample Correction

Since there are no labeled data during the self-supervised pretraining process, we use the positive sample pair with the highest similarity in the feature space as the calibration benchmark based on the obtained geography feature space joint distance. For the self-supervised contrastive learning model, the positive sample pairs are the different augmentations of the same image, and the highest similarity in the feature space implies that the model captures the invariant features of these pairs to the maximum extent [25,33,65].

Specifically, we regard the negative sample with the smallest difference in the geography-feature space joint distance from the calibration benchmark as the positive sample, removing negative samples that contained the same ground object as the anchor sample to mitigate class confusion.

(3) Feature Extraction and Contrastive Loss Calculation

Finally, we input the corrected negative and positive samples into the feature extractor to obtain the features and calculate the contrastive loss (the loss function for self-supervised contrastive learning and also the minimization objective function for updating the parameters of the contrast learning model) to update the model parameters. In this part, we adopt InfoNCE [66] as the contrastive loss, which is defined as follows:

$$L = -log \frac{exp(sim(f_i, f_j)/\tau)}{\Sigma_{k=1, k \neq i}^{N} exp(sim(f_i, f_k)/\tau)},\tag{5}$$

where $f_i$ and $f_j$ denote the positive sample features, $sim()$ denotes the calculation of the similarity between the two image features, and $\tau$ denotes the temperature hyperparameter.

### 2.2.3. Multi-Scale Feature Joint Land Cover Classification Fine-Tuning

Multi-scale feature joint land cover classification fine-tuning aims to integrate the features of different scales of images obtained by the self-supervised contrastive learning model, enhance the ability of the model to capture ground objects of different scales, and further improve the performance of downstream land cover classification tasks.

Specifically, in the fine-tuning stage of self-supervised contrastive learning, we utilize the single-layer convolutional network to obtain different scale features of the three middle layers of the feature extractor, and align these features with the dimensions of the features in the last layer of the feature extractor. Then, we input the obtained features of different scales into the single-layer convolutional network for integration, and the obtained multi-scale features are fed into the land cover decoder to obtain the land cover classification results of the image. The detailed structure of the MFFS is shown in Figure 3.

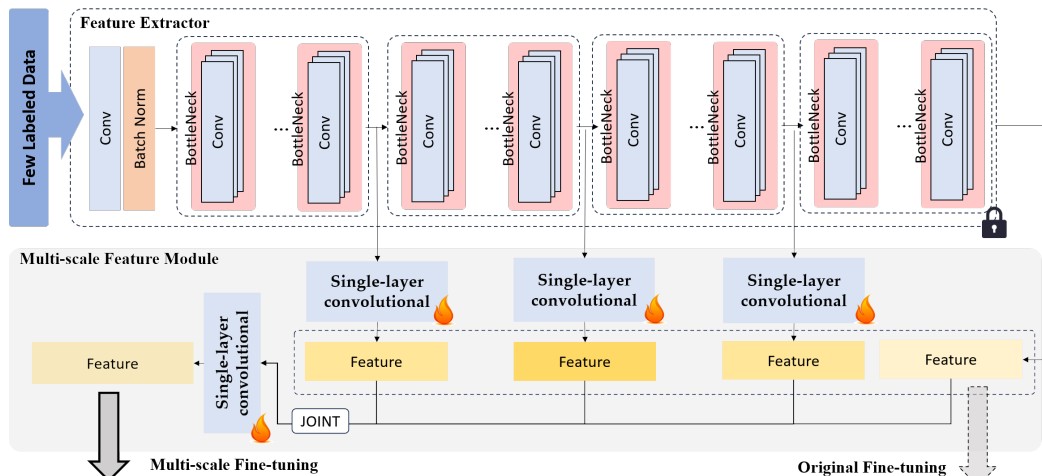

**Figure 3.** Schematic of the detailed structure of Mutil-scale Feature joint Land cover classification Fine-tuning Strategy (MFFS), where the "lock" symbol in the lower right corner of the model indicates freezing network parameters and the "fire" symbol indicates updating network parameters. Different from only using the single-scale features obtained from the last layer of the feature extractor for fine-tuning, we first integrate the different scale features obtained from the feature extractor by using the added single-layer convolutional and then use the obtained multi-scale joint features for fine-tuning.

## 3. Experiments

### 3.1. Dataset Description

We evaluate the performance of the proposed GFCNet on three land cover classification datasets: Five Billion Pixel [11], LoveDA Urban, and LoveDA Rural [67].

The Five-Billion-Pixel dataset is derived from the Gaofen-2 satellite, and the spatial resolution is 4 m, which adds more ground object categories to the GID dataset [10] and covers an area of more than 50,000 km$^2$ in China. The LoveDA Urban and LoveDA Rural datasets have a spatial resolution of 0.3 m and cover urban and rural areas in three regions of Nanjing, Changzhou, and Wuhan, China [67]. The dividing of the training and test data sets is consistent with the original dataset. Table 1 shows more details of the Five-Billion-Pixel, LoveDA Urban, and LoveDA Rural datasets.

**Table 1.** The detail informations of Five Billion Pixel, LovaDA Urban, and LoveDA Rural.

| Dataset | Five-Billion-Pixel | LoveDA Urban | LoveDA Rural |
|---|---|---|---|
| Year | 2023 | 2021 | 2021 |
| Resolution | 4 m | 0.3 m | 0.3 m |
| Area | >50,000 km$^2$ | 245.75 km$^2$ | 289.41 km$^2$ |
| Class Number | 24 | 7 | 7 |
| Crop Size | $512 \times 512$ | $256 \times 256$ | $256 \times 256$ |
| Amount of Data for SSL Pretraining (with no label) | 25,200 | 18,496 | 21,856 |
| Amount of Data for Fine-tuning (1% of the pretraining data) | 252 | 184 | 218 |
| Amount of Data for Testing | 6300 | 10,832 | 15,872 |

### 3.2. Baselines and Metric

In order to evaluate the performance of the proposed GFCNet, we use seven self-supervised contrastive learning methods with different positive and negative sample construction strategies as baselines:

SimCLR [21] and MoCo v2 [36] use a heuristic positive and negative sample construction method, which treats the different augmentations of the same image as positive samples, and treats the augmentations of different images as negative samples.

Barlow Twins [48] and BYOL [49] only treat different augmentations of the same image as positive samples and do not construct negative samples during contrastive learning.

PCL [38] introduces a clustering method when constructing positive and negative samples.

FALSE [25] considers reducing negative samples that contain the same ground object with the discriminative ability of the model.

DenseCL [42] adds a dense contrastive module to improve the ability of the model to capture objects of different scales.

In addition, we refer to the relevant land cover classification methods [9–11] to adopt Overall Accuracy (OA), Kappa and mean Intersection-over-Union (mIoU) to quantitatively evaluate the performance of the self-supervised contrastive learning model on the downstream land cover classification task, where OA is the overall accuracy of the model prediction, which can be calculated with the following equation:

$$OA = \frac{TP}{N}, \tag{6}$$

where $TP$ denotes the number of true positives, which is the number of pixels correctly predicted by the model, and $N$ denotes the total number of pixels in the test dataset.

Kappa can be calculated by the following equation:

$$Kappa = \frac{OA - p_e}{1 - p_e}, \tag{7}$$

$$p_e = \frac{\Sigma_{i=1}^n g_i \cdot p_i}{N \cdot N}, \tag{8}$$

where $g_i$ denotes the number of pixels in class $i$ of ground truth, and $p_i$ denotes the number of pixels in class $i$ obtained by the model.

Mean Intersection-over-Union (mIoU) can be calculated by the following equation:

$$mIoU = \Sigma_{i=1}^N IoU_i, \tag{9}$$

$$IoU_i = \frac{TP}{TP + FN + FP}, \tag{10}$$

where $TP$ denotes the number of true positives, $FN$ denotes the number of false negatives, and $FP$ denotes the number of false positives.

### 3.3. Implementation Details

All experiments are performed on NVIDIA RTX A6000, the software environment of Python 3.7 with PyTorch as the deep learning framework.

In the self-supervised contrastive pretraining stage, for all self-supervised contrastive learning methods, we uniformly train 200 epochs, and the batch size is set to 256. For the self-supervised contrastive learning methods that use InfoNCE as the contrastive loss, the temperature hyperparameter of infoNCE is set to 0.5, and the settings of the other baseline methods are adopted as the default given in the original paper.

In the land cover classification fine-tuning stage, for all the self-supervised contrastive learning methods, we freeze the weights of the feature extractors, train the land cover classification decoder for 150 epochs, and set the batch size to 16.

### 3.4. Experiment Results

3.4.1. Performance Analysis

The quantitative results of the performance analysis are shown in Table 2. The experimental results show that compared with seven self-supervised contrastive learning methods with different positive and negative sample construction methods, our proposed GFCNet achieves the best results on three land cover classification datasets of Five Billion Pixel, LoveDA Urabn, and LoveDA Rural. In addition, on the LoveDA Rural dataset,

the proposed GFCNet improves Kappa by 5.7% and mIoU by 4.85% compared with SimCLR which uses the classical positive and negative sample construction strategy, and improves Kappa by 3.87% and mIoU by 1.54% compared with the best baseline DenseCL.

**Table 2.** Quantitative comparison results with seven self-supervised contrastive learning baseline methods on Five Billion Pixel, LoveDA Urban, and LoveDA Rural. Bold numbers represent the highest value under the metric, and underlined numbers represent the second-highest value under the metric.

| Method | Five Billion Pixel | | | LoveDA Urban | | | LoveDA Rural | | |
|---|---|---|---|---|---|---|---|---|---|
| | OA | Kappa | mIoU | OA | Kappa | mIoU | OA | Kappa | mIoU |
| SimCLR | 64.31 | 55.80 | 21.34 | 40.92 | 27.51 | 32.10 | 61.32 | 45.65 | 37.83 |
| MoCo v2 | 53.03 | 40.89 | 14.13 | 40.30 | 27.32 | 32.72 | 60.66 | 45.25 | 38.54 |
| Barlow Twins | 58.66 | 48.01 | 17.90 | 41.19 | 29.36 | 32.96 | 57.10 | 41.61 | 37.01 |
| BYOL | 64.10 | 55.47 | 21.17 | 32.67 | 15.71 | 24.14 | 59.86 | 42.70 | 37.42 |
| PCL | 57.98 | 47.58 | 17.29 | 40.05 | 27.12 | 33.08 | 62.98 | 47.26 | 40.24 |
| FALSE | 64.88 | 56.69 | 21.41 | 41.22 | 27.85 | 32.67 | 62.41 | 46.43 | 40.96 |
| DenseCL | 51.35 | 39.03 | 13.43 | 36.62 | 23.34 | 30.21 | 63.04 | 47.48 | 41.14 |
| **GFCNet(Ours)** | **65.44** | **57.30** | **21.84** | **42.36** | **29.48** | **34.06** | **65.61** | **51.35** | **42.68** |

Figure 4 shows the land cover classification visualization results of the proposed GFCNet and seven baseline methods on three datasets. The experimental result shows that the proposed GFCNet effectively reduces class confusion compared to seven baseline methods, and the ground object mixing in the classification results is significantly eliminated. In addition, we note that the proposed GFCNet can more effectively capture small-scale ground objects in remote sensing images, such as tiny oases existing within a large body of water, and small-scale buildings in the image.

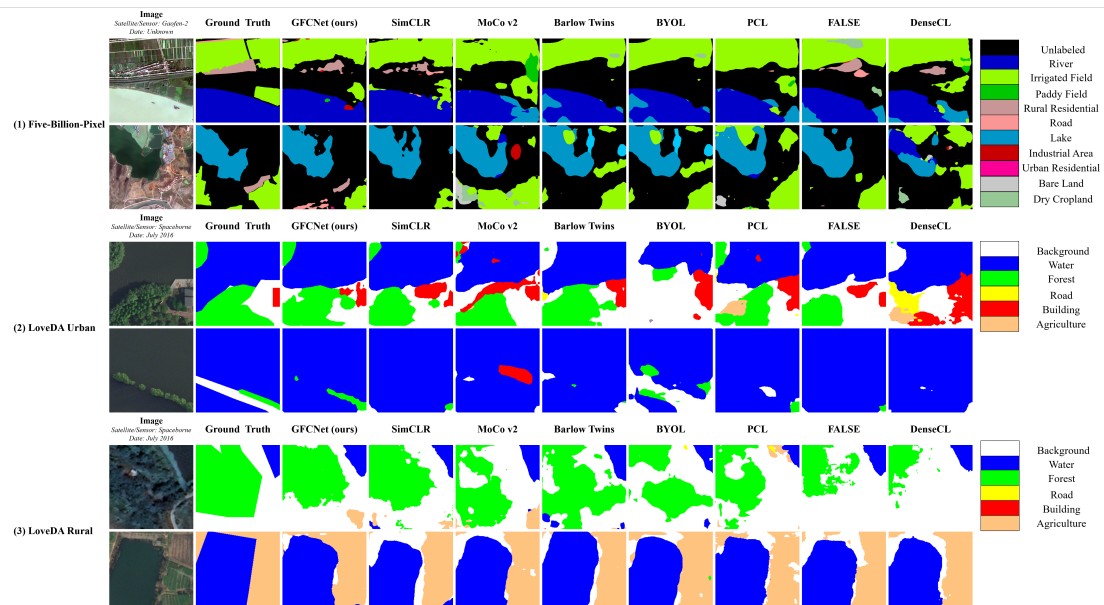

**Figure 4.** Visualization Results of the Proposed GFCNet and seven baseline methods for land cover classification on three datasets of Five Billion Pixel, LoveDA Urban and LoveDA Rural.

### 3.4.2. Ablation Study

To evaluate the impact of the Geography Feature space joint negative sample Correction Strategy (GFCS) and the Multi-scale Feature join Fine-tuning Strategy (MFFS) on the model performance, we conducted an ablation study on the proposed GFCNet. In Table 3, SimCLR represents a self-supervised contrastive learning method using the original positive and negative sample construction strategy, SimCLR + FNSD represents the SimCLR

with False Negative sample Self-Determination (FNSD) strategy [25], the FNSD is a negative sample correction strategy that utilizes the self-determination ability of the model and the feature space relationship of an image, and SimCLR + GFCS represents the SimCLR with Geography Feature space joint negative sample Correction Strategy (GFCS).

**Table 3.** The experimental results of ablation study on Five Billion Pixel, LoveDA Urban, and LoveDA Rural datasets.

| Method | FNSD | GFCS | MFFS | Five-Billion-Pixel | | LoveDA Urban | | LoveDA Rural | |
|---|---|---|---|---|---|---|---|---|---|
| | | | | OA | Kappa | OA | Kappa | OA | Kappa |
| SimCLR | | | | 64.31 | 55.80 | 40.92 | 27.51 | 61.32 | 45.65 |
| SimCLR + FNSD | ✓ | | | 64.88 | 56.69 | 41.22 | 27.85 | 62.41 | 46.43 |
| SimCLR + GFCS | | ✓ | | 65.20 | 57.08 | 41.91 | 29.65 | 65.27 | 51.02 |
| GFCNet | | ✓ | ✓ | 65.44 | 57.30 | 42.36 | 29.48 | 65.61 | 51.35 |

The experimental results in Table 3 show that the proposed GFCS effectively improves the performance of the self-supervised contrastive learning model on the land cover classification task compared to FNSD without MFFS, and that the performance of the model is further improved with the MMFS. This indicates that the geospatial relationship information of the image can provide additional information for the model to eliminate the class confusion caused by the negative samples containing the same ground objects in the self-supervised contrastive learning model.

In addition, to further verify the performance of the Multi-scale Feature join Fine-tuning Strategy (MFFS), we combine LoveDA Urban and LoveDA Rural datasets to increase the difficulty of multi-scale feature extraction, and compare with separate LoveDA Urban and LoveDA Rural datasets to verify the ability of the proposed method to capture multi-scale objects. The experimental results are shown in Table 4.

**Table 4.** Comparative performance experimental of Multi-scale Feature join Fine-tuning Strategy (MFFS) on LoveDA Urban, LoveDA Rural, and LoveDA (Urban + Rural) datasets.

| Method | Original Fine-Tuning | MFFS | LoveDA Urban | | LoveDA Rural | | LoveDA (Urban + Rural) | |
|---|---|---|---|---|---|---|---|---|
| | | | OA | Kappa | OA | Kappa | OA | Kappa |
| GFCNet | ✓ | | 41.91 | 29.65 | 65.27 | 51.02 | 56.16 | 41.81 |
| | | ✓ | 42.36 | 29.48 | 65.61 | 51.35 | 57.16 | 43.95 |

The experimental results illustrate that compared with the original fine-tuning method, the proposed Multi-scale Feature join Fine-tuning Strategy (MFFS) obtains performance improvement of land cover classification tasks on both LoveDA Urban, LoveDA Rural, and LoveDA (Urban + Rural) which combines LoveDA Urban and LoveDA Rural dataset. Moreover, the performance improvement of the proposed MFFS on the LoveDA (Urban + Rural) dataset is higher than that on the LoveDA Urban and LoveDA Rural datasets, which indicates that MFFS effectively improves the ability of the model to capture multi-scale ground objects.

### 3.4.3. Domain Adaptation Analysis

One of the goals of self-supervised contrastive pretraining is to obtain a powerful image feature extractor that can be fine-tuned for out-of-domain data [20,23]. For land cover classification tasks, the powerful image feature extractor can help reduce the dependence on high-quality labeled data. In this part, we conducted two experiments for domain adaptation analysis of the proposed GFCNet. The first domain adaptation analysis uses LoveDA Urban and LoveDA Rural datasets, and the second domain adaptation analysis

uses the Five-Billion-Pixel dataset and the LoveDA (Urban + Rural) dataset which combines LoveDA Urban and LoveDA Rural datasets.

Specifically, we conducted two sets of experiments on the LoveDA Urban and LoveDA Rural datasets. The first set of experiments used a pretraining dataset that is consistent with the fine-tuning dataset, and utilized the test data corresponding to the fine-tuning dataset to evaluate the performance of the models. The second set of experiments used a pretraining dataset that is inconsistent with the fine-tuned dataset, and again utilized the test data corresponding to the fine-tuned dataset to evaluate the performance of the model. In addition, we also compared GFCNet with SimCLR, which uses the original positive and negative sample construction strategy, and FALSE, a contrastive learning model that removes false negative samples. The experimental results are shown in Table 5. We also conducted the above two sets of experiments on LoveDA (Urban + Rural) and Five-Billion-Pixel datasets, and the experimental results are shown in Table 6.

**Table 5.** The experimental results of domain adaptation analysis on LoveDA Urban and LoveDA Rural datasets.

| Fine-Tuning and Validation Dataset | Pretraining Dataset | SimCLR | | FALSE | | GFCNet | |
|---|---|---|---|---|---|---|---|
| | | OA | Kappa | OA | Kappa | OA | Kappa |
| LoveDA Urban | LoveDA Urban | 40.92 | 27.51 | 41.22 | 27.85 | 42.36 | 29.48 |
| | LoveDA Rural | 40.26 | 27.41 | 41.20 | 27.64 | 42.09 | 29.56 |
| LoveDA Rural | LoveDA Rural | 61.32 | 45.65 | 62.41 | 46.43 | 65.61 | 51.35 |
| | LoveDA Urban | 58.33 | 41.40 | 60.92 | 45.29 | 61.00 | 45.34 |

**Table 6.** The experimental results of domain adaptation analysis on Five-Billion-Pixel and LoveDA (Urban + Rural) datasets.

| Fine-Tuning and Validation Dataset | Pretraining Dataset | SimCLR | | FALSE | | GFCNet | |
|---|---|---|---|---|---|---|---|
| | | OA | Kappa | OA | Kappa | OA | Kappa |
| LoveDA (Urban + Rural) | LoveDA (Urban + Rural) | 55.47 | 41.21 | 55.65 | 41.63 | 57.16 | 43.95 |
| | Five-Billion-Pixel | 54.70 | 40.28 | 55.36 | 41.36 | 56.52 | 43.48 |
| Five Billion Pixel | Five Billion Pixel | 64.31 | 55.80 | 64.88 | 56.69 | 65.44 | 57.30 |
| | LoveDA (Urban + Rural) | 64.19 | 55.77 | 64.71 | 56.32 | 65.40 | 57.37 |

The experimental results indicate that SimCLR, FALSE, and GFCNet show performance degradation in the land cover classification task when the pretraining dataset and the fine-tuning dataset are inconsistent compared with the pretraining dataset and the fine-tuning dataset are consistent.

For the first domain adaptation analysis experiment, when the test data are obtained from the LoveDA Urban dataset, the land cover classification performance of GFCNet pretrained with the LoveDA Rural dataset still exceeds SimCLR and FALSE pretrained with the LoveDA Urban dataset, which indicates that GFCNet has a certain domain adaptation ability. When the test data are obtained the LoveDA Rural dataset, the performance of SimCLR, FALSE, and GFCNet pretrained with the LoveDA Urban dataset all show a significant performance decline, but the land cover classification performance of GFCNet is still higher than SimCLR and FALSE under the same conditions.

For the second domain adaptation analysis experiment, when the test data are obtained from LoveDA (Urban + Rural), the performance of GFCNet pretrained with the Five-Billion-Pixel datasets also exceeds SimCLR and FALSE pretrained with the LoveDA (Urban + Rural) dataset, which again shows that GFCNet has a certain domain adaptation ability. In addition, when the test data are Five Billion Pixel, the performance degradation of GFCNet pretrained with the LoveDA (Urban + Rural) dataset is lower than that of SimCLR

and FALSE. SimCLR is decreased by 0.12% on OA, and FALSE is decreased by 0.17% on OA. In contrast, GFCNet is only decreased by 0.04% on OA.

From the perspective of model construction, due to the comprehensive consideration of the geography space and feature space relationships of images, GFCNet using Geography Feature space joint negative sample Correction Strategy (GFCS) captures higher-quality image features than SimCLR and FALSE in the self-supervised contrastive learning pretraining stage, which comes from the higher-quality positive and negative samples constructed by GFCS. In addition, the Multi-scale Feature join Fine-tuning Strategy (MFFS) adopted in the fine-tuning stage of GFCNet also enhances the ability of the model to capture the features of ground objects at different scales to a certain extent. These two factors together make GFCNet have better domain adaptation ability than SimCLR and FALSE.

### 3.4.4. Visualization of Negative Sample Correction

In order to explore the impact of the Geography Feature space joint negative sample Correction Strategy (GFCS) on negative sample correction, we visualize and analyze the negative samples screened by GFCS and compared with the False Negative Self-Determination (FNSD) strategy.

Specifically, after the pretraining of the model, we pick an image as an anchor sample and input this image together with other images in the dataset into the model to obtain the image features. For FNSD, we calculate the feature similarity between the anchor sample and other images, and then sort them to obtain the five images with the highest similarity as the false negative sample obtained from FNSD. For the proposed GFCS, we calculate the geography feature space joint distance between the anchor sample and other images, and then sort them to obtain the five images with the highest geography feature space joint distance as the false negative sample obtained from GFCS. The experimental results are shown in Figure 5.

The experimental results show that compared to FNSD, GFCS prefers to select images that are closer to the anchor sample in both geospatial and visual features as negative samples that contain the same ground object with the anchor sample. Especially when there are large continuous ground objects in the image, the geospatial relationship of the image can effectively assist the self-supervised contrastive learning model to screen the negative samples and mitigate class confusion.

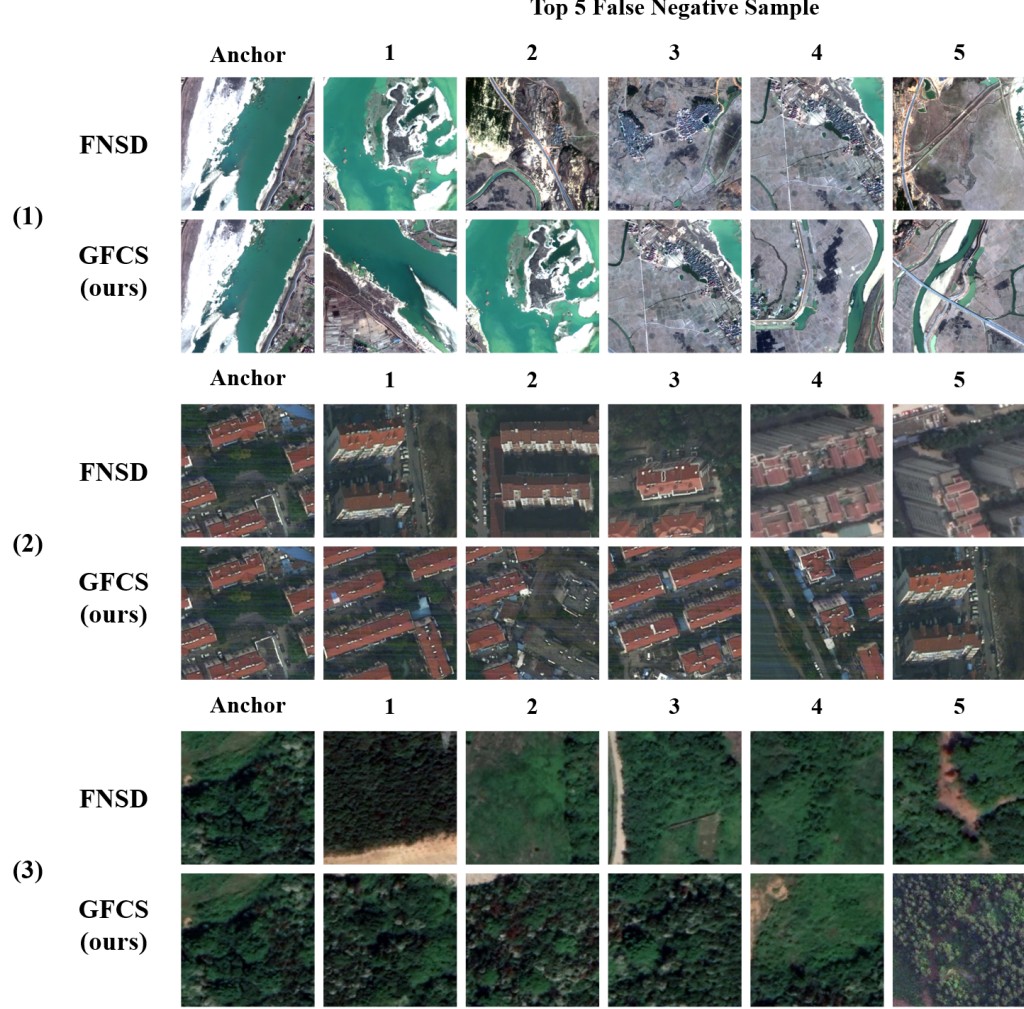

**Figure 5.** Visualization results of negative sample correction, where the anchor sample and false negative samples in (1) are from the Five-Billion-Pixel dataset, the anchor sample and false negative samples in (2) are from the LoveDA Urban dataset, and the anchor samples and false negative samples in (3) are from the LoveDA Rural dataset.

## 4. Discussion

In this study, we propose a Geography Feature space joint negative sample Correction Strategy (GFCS) and a Multi-scale Feature joint Fine-tuning Strategy (MFFS) of a self-supervised learning model for land cover classification. The GFCS effectively mitigates class confusion, and the MFFS improves the ability of the models to capture ground objects at different scales.

We observe that the proposed GFCNet has a higher performance improvement compared to the baseline method in processing remote sensing images that contains a large range of homogeneous ground objects. Among the three land cover classification datasets used in our experiments, the LoveDA Rural dataset sampled from rural areas; it contains a large number of large-scale farmland and forests, which means that image patches taken from the neighboring areas have a higher probability of containing the same ground object, whereas the proposed GFCNet introduces geospatial relationships of images, which provides effective information for the model to obtain negative samples that are geospatially neighboring and contain the same ground object as the anchor sample. Further ablation studies reaffirm this; under the condition of removing the MFFS, the proposed GFCS achieves greater gains on the LoveDA Rural dataset compared to FNSD, which only considers the feature space relationship of images.

In addition, we also notice that the performance improvement of the model tends to be marginal with the inclusion of MFFS, which is due to the fact that the contrastive method used is still instance-level contrastive. The instance-level contrastive limits the ability of the model to capture features of different scales [68].

Although the proposed GFCS cannot completely screen out all negative samples that contain the same ground object as the anchor samples, it can effectively mitigate the ground object class confusion and make the model have strong domain adaptation ability. For example, the model pretrained by the proposed GFCNet on the LoveDA Rural dataset can be used for feature extraction of the LoveDA Urban dataset after fine-tuning, and the overall accuracy of the model decreases by only 0.27%.

## 5. Conclusions

In this paper, we propose a contrastive learning network with Geography Feature space joint negative sample Correction (GFCNet) for land cover classification. To address the serious ground object class confusion, we develop a Geography Feature space joint negative sample Correction Strategy (GFCS), which effectively mitigates the risk of negative samples containing the same ground object as the anchor sample. In order to improve the ability of the model to capture different scales of ground objects, we adopt a Multi-scale Feature joint Fine-tuning Strategy (MFFS) to integrate different scale features obtained by the self-supervised contrastive learning model for land cover classification tasks. We validate the effectiveness of the proposed GFCNet on three public land cover classification datasets. The performance of the GFCNet surpassed seven self-supervised contrastive learning baseline methods.

Although this approach is effective for screening negative samples that contain the same ground object as the anchor sample and are geospatially closer to the anchor sample, it also means that it is difficult for the model to screen negative samples that are geospatially farther away from the anchor sample but contain the same ground object. Therefore, ways to make the model have a longer range negative sample correction ability in geography space and to better integrate the geospatial information of remote sensing images and the feature extraction ability of the model are worthy of research in the future.

**Author Contributions:** Conceptualization, methodology, software, and writing—original draft preparation: Z.Z.; supervision: H.L., Y.Z. and C.T.; software and validation: W.J.; data curation: W.J. and C.T.; writing—reviewing and editing: Y.Z. All authors have read and agreed to the published version of the manuscript.

**Funding:** The authors gratefully acknowledge the financial support provided by the Major Program Project of Xiangjiang Laboratory (No. 22XJ01010).

**Data Availability Statement:** Data associated with this research are available online. The Five-Billion-Pixel dataset is available at https://x-ytong.github.io/project/Five-Billion-Pixels.html (accessed on 7 April 2023). The LoveDA Urban and LoveDA Rural dataset is available at https://github.com/Junjue-Wang/LoveDA (accessed on 17 October 2021).

**Conflicts of Interest:** The authors declare no conflict of interest.

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
