# Peer review of "GFCNet: Contrastive Learning Network with Geography Feature Space Joint Negative Sample Correction for Land Cover Classification"

_remotesensing, doi:10.3390/rs15205056_

Round 1
Reviewer 1 Report
In this study, the authors proposed a geo-feature space joint negative sample correction (GFCNet) contrast learning network, used for land cover classification. In order to solve the classification confusion problem of existing contrastive learning, we propose a geo-feature space joint negative sample correction strategy (GFCS), which integrates the geospatial and feature space relationships of different images to construct negative samples, reducing the risk of negative samples containing the same object. In order to improve the ability of the model to capture the features of ground objects at different scales, the Multi-scale Feature joint Fine-tuning Strategy (MFFS) is proposed. The experiments are carried out on three data sets, and the results of SOTA are achieved. Overall, this work presents a geo-specific self-supervised approach to contrastive learning that is sufficiently innovative and experimental to merit publication. Here are some suggestions:
1. In the abstract, the authors show the largest improvement over the comparison method. It is tempting to think that the improvement is much higher than SOTA. In fact, the improvement is not so obvious compared to FALSE.
2. The comparison diagram between the positive and negative samples constructed by the proposed method and the positive and negative samples constructed by simCRL and False can be added to facilitate the intuitive understanding of the advantages of the proposed positive and negative sample construction method.
Reviewer 2 Report
In this manuscript, the authors introduce GFCNet, a self-supervised deep learning model designed for land cover mapping. The article is concise, clearly written, and well-structured, offering a potentially significant advancement in the field of land cover mapping using self-supervised learning.
The primary strengths of the paper lie in its clever methodology and clarity of presentation. The proposed GFCNet, if validated appropriately, could significantly reduce the time and resources needed for labeling data, which is needed for supervised classification methods, without a significant reduction in accuracy.
However, there are key issues that need to be addressed. These primarily relate to the divergence between methods and terminologies commonly used by the machine learning/AI communities and those accepted by the “traditional” (or pre-deep learning) remote sensing community. Specifically, the paper adopts accuracy metrics and terminologies that, while may be standard in machine learning literature, are not widely accepted or are even considered flawed in remote sensing, as evident by years of research on map accuracy assessments. Without similar research on “good practices” for the robust assessment of deep learning models for land cover mapping applications, it is hard to determine the robustness and the results. Furthermore, it can create a barrier to the paper’s acceptance and applicability within the broader remote sensing community.
However, I believe these issues can be addressed and offer suggestions for each section below.
Background
The Background section is well-written and effectively introduces the central problem and objectives of the study. However, several areas could benefit from further elaboration to make the paper more accessible and informative.
Terminology:
- The paper uses technical terms that are primarily used in the machine learning/AI communities. As Remote Sensing covers a wide range of topics, it is possible that many readers may not be familiar with these terms. Therefore, providing brief definitions or explanations would be beneficial to aid in the understanding of the paper.
- Certain words and phrases are used differently in the remote sensing community than in the ML/AI communities. For instance, in traditional map accuracy assessments, a “sample” is derived from a population (such as all the pixels on a map) and contains many individual “sample units”. Although the fields of remote sensing and ML/AI are rapidly converging, it's crucial for authors to consider their intended audience to avoid any confusion while describing their research.
Context
The paper contends that the most advanced remote sensing methods are currently based on Deep Learning. While this may be true and is certainly the direction the community is headed, the authors provide no context as to why there has been a semi-recent paradigm shift in this direction. It would be valuable to include a brief comparison with traditional remote sensing methodologies to better understand the advantages of Deep Learning, specifically self-supervised methods, over traditional classification techniques.
References:
- The literature review is robust for the specific topic of the paper, which is self-supervised contrastive learning, but lacking in the larger context of land use and land cover mapping. The introduction could be strengthened by including key references that trace the evolution of land cover mapping methodologies, leading to the use of Deep Learning and specifically self-supervised learning. This would provide a better justification for the proposed GFCNet model and would provide a more comprehensive view of the field's trajectory.
Research Design
The research design is generally appropriate for the study's objectives and aligns well with the methodologies commonly employed in the field. However, there are concerns related to the accuracy assessment that warrant attention.
The paper uses Overall Accuracy, Kappa, and mean Intersection-over-Union statistics as the primary metrics for evaluating the model performance. While these metrics may be common for evaluating a non-spatial model, their appropriateness for remote sensing applications has been questioned. One of the original authors of multiple Kappa statistics even published a paper titled "Death to Kappa," highlighting that these indices are not well-suited for remote sensing applications, going as far as saying they are “useless, misleading and/or flawed” (Pontius and Millones, 2011).
The remote sensing community has established certain "good practices" for accuracy estimation, prioritizing probability-based sampling approaches. These approaches were developed to address specific issues relevant to geospatial applications, such as spatial autocorrelation, and provide accurate and statistically defensible accuracy metrics with well-characterized uncertainties. (i.e., Stehman 1997, Olofsson et al. 2014, Espejo et al. 2020). As it goes against widely accepted “good practice guidelines” in the land cover mapping community, it is therefore critical for the authors to address and justify the usage of their chosen performance metrics for evaluating GFCNet. Failure to do so could undermine the perceived validity of the results, even if the chosen metrics are technically appropriate for this specific case.
Methods
As previously mentioned, the methods described in the paper are presented with excellent clarity. The figures and equations are well-crafted, and the writing follows a consistent flow that is generally easy to follow. However, it is worth noting that the paper will be very difficult to read for those without advanced knowledge of DL methods. Since Remote Sensing has a broad readership, I suggest providing brief descriptions of any technical terms specific to the discipline to improve the paper's readability.
Results
The results are presented clearly and are well structured. I suggest providing more details to the “Image” column in Figure 3 (e.g., date, satellite/sensor).
As previously noted, the Kappa statistic has been described as “useless, misleading and/or flawed” for remote sensing applications (Pontius and Millones, 2021, Abstract). Similar insights have also been made about the use of Overall Accuracy. Maxwell et al. (2021) provides an excellent summary of how deep learning papers diverge from traditional remote sensing accuracy assessments and the problems that could be introduced. It may be that the methods and results accurately support the conclusion that GFCNet outperforms similar models, but it is hard to be certain without more details about the chosen performance metrics. Providing more details on the performance metrics and test datasets, including a clear justification about their robustness in the context of this manuscript, would avoid confusion and ensure the results support the conclusions.
References
Espejo, A. B., Federici, S., Green, C., Olofsson, P., Sánchez, M. J. S., Waterworth, R., ... & Herold, M. (2020). Integrating remote-sensing and ground-based observations for estimation of emissions and removals of greenhouse gases in forests: Methods and Guidance from the Global Forest Observations Initiative. Edition 3.0. GFOI.
Maxwell, A. E., Warner, T. A., & Guillén, L. A. (2021). Accuracy assessment in convolutional neural network-based deep learning remote sensing studies—Part 2: Recommendations and best practices. Remote Sensing, 13(13), 2591.
Pontius Jr, R. G., & Millones, M. (2011). Death to Kappa: birth of quantity disagreement and allocation disagreement for accuracy assessment. International Journal of Remote Sensing, 32(15), 4407-4429.
Stehman, S. V. (1997). Selecting and interpreting measures of thematic classification accuracy. Remote Sensing of Environment, 62(1), 77–89.
Reviewer 3 Report
This paper introduces a contrastive learning network with Geography-Feature space joint negative sample Correction, which is driven by a large amount of unlabeled data. To solve the problem of class confusion and improve the ability of the model to capture features of objects of different scales, a Geography-Feature space joint negative sample Correction Strategy is proposed. At the same time, a Multi-scale Feature joint Fine-tuning Strategy is adopted, and the experimental results show that it achieves relatively advanced performance. While this work may hold research implications, the innovations it offers remain somewhat limited. Furthermore, specific comments and suggestions are as follows:
1.In line 153, you mentioned that existing self-supervised contrastive learning models only utilize simple single-scale features. As you mentioned earlier, there are methods attempting to improve the utilization of different-scale objects, but they primarily focus on the pre-training phase. From my understanding, incorporating multi-scale features is not a new idea and can be easily implemented. Could you please explain in detail how your approach differs from previous methods and why it can be considered innovative?
2. The Figure 1 in the manuscript is not drawn clearly enough as an overall framework, making it difficult to see the connection between the first and second stages. In particular, the name of the second stage figure is "Multi-scale Feature Joint Land Cover Classification Fine-tuning ", but from the figure, we cannot see the idea of integrating multi-scale features, nor can we see the fine-tuning strategy. All that can be seen is a network for land cover classification.
3.The focus of the article should be on the innovative method, where a diagram is missing in Section 2.2.2 for the stage of "Geography-Feature Space Joint Negative Sample Correction Contrastive Pretraining", making it difficult to understand the workings of GFCS intuitively.
4.The diagram of the MFFS structure in the Figure 2 does not show the fine-tuning work it performs, especially the fine-tuning effect and principle of multi-layer features, making it difficult to see its function.
5.The discussion of the impact of GFCS on negative sample correction and its visualization in Section 3.4.4 should be placed in the section introducing this method to demonstrate its effectiveness.、
6. There are some deficiencies in the writing and logic of the article. It is recommended that the author learn from other articles on land cover classification, such as https://www.sciencedirect.com/science/article/abs/pii/S0924271621003270 and https://www.sciencedirect.com/science/article/pii/S0924271622003264.
7.The description used in the abstract is "achieved a maximum improvement of 5.7% in Kappa and 4.85% in mIoU compared to the baseline of seven self-supervised contrastive learning methods", but the actual comparison was with the SimCLR baseline. Please make this description more rigorous.
There are some linguistic and logical flaws in this article, and some paragraphs have poor readability. It is recommended that the author make improvements.
Reviewer 4 Report
1. the LoveDA dataset can combine urban and rural areas in the semantic segmentation task, which increases the difficulty of multi-scale feature extraction, and will have a fuller representation of the adaptability of this paper's method. It is recommended that the authors combine these two methods to verify that the proposed method has a better ability to capture multi-scale objects
2. in domain adaptation analysis, if the pre-training dataset is Five Billion Pixels, then fine-tune LoveDA. what happens if the pre-training dataset is LoveDA and the fine-tuning dataset is Five Billion Pixels? The authors are advised to explore this further.
Reviewer 5 Report
The manuscript proposed a contrastive learning network to accomplish the task of land cover classification, built the GFCS module for better learning of negative samples, and made the model better at integrating multi-scale features by building the MFFS module, which is innovative. However, there are some detailed parts in the paper that need further confirmation.
1、 On the second page, lines 84-87, the dataset used for the experiments and the precision metrics are not reasonably summarized in the introduction.
2、 In Related Work section, the review on land cover classification is inadequate and it is recommended that more relevant literature be added.
3、 The structure of the multi-scale feature in Figure 2 is not clear enough. Single-layer convolutional is not reflected in the figure. It is suggested to add the relevant content in the figure.
4、 A detailed description of the Implementation Details is missing in Section 3, and it is recommended that it be added. For example the version of python and deep learning framework.
5、 In subsection3.4.3, it has been proved the proposed methods has better domain adaptive ability. Please analyze the reason for the method having better domain adaptive ability in terms of the construction of the model.
Round 2
Reviewer 3 Report
The authors have revised the manuscript according to the reviewers' comments, and the results of the manuscript have been improved.
All my concerns have been addressed, and I have no problem recommending it for publication.
Reviewer 4 Report
The concerns that I proposed are all properly addressed. Now I'm sure that this revised manuscript can be accepted for publication.